# Understanding the determinants of treated bed net use in Ethiopia: A machine learning classification approach using PMA Ethiopia 2023 survey data

Abraham Keffale Mengistu[ID]*

Department of Health Informatics, College of Medicine Health Science, Debre Markos University, Debre Markos, Ethiopia

* abreham_keffale@dmu.edu.et

## Abstract

### Introduction

Malaria remains a significant public health challenge in Ethiopia, with over 7.3 million cases and 1,157 deaths reported between January 1 and October 20, 2024. Despite extensive distribution campaigns, 35% of insecticide-treated nets (ITNs) remain underutilized, hindering malaria control efforts. Traditional statistical approaches have identified socioeconomic and demographic factors as predictors of ITN use, but often fail to capture complex, nonlinear interactions. This study applies machine learning to identify non-apparent factors of ITN utilization and investigates its performance in prediction as compared to traditional logistic regression.

### Methods

This study applied ML models, including Random Forest, XGBoost, and Gradient Boosting, to predict ITN utilization using the 2023 Performance Monitoring for Action (PMA) Ethiopia dataset, a nationally representative survey of 9,763 households. The dataset included 18 variables: region, household size, wealth quintile, and housing conditions. Model performance was evaluated using accuracy, precision, recall, F1-score, and AUC-ROC. The values of SHAP (Shapley Additive Explanations) were used to interpret feature importance and interaction effects.

### Results

Random Forest and XGBoost outperformed traditional logistic regression, achieving AUC scores of 0.89(0.91 after optimization) and 0.88, respectively. Key determinants of ITN utilization included geographic region, household size, wealth quintile, and maternal education. Nonlinear interactions, such as the moderating effect of maternal education on income-related barriers, were identified. Regional disparities were

**Data availability statement:** The datasets used and/or analyzed during the current study are available on the PMA website (https://doi.org/10.34976/k8hq-b666).

**Funding:** The author(s) received no specific funding for this work.

**Competing interests:** The Author has no competing interests

**Abbreviations:** AUC-ROC, Area Under the Receiver Operating Characteristic Curve; CFR, Case Fatality Rate; EAs, Enumeration Areas; HIV, Human Immunodeficiency Virus; ICU, Intensive Care Unit; ITNs, Insecticide-Treated Nets; KNN, K-Nearest Neighbors; ML, Machine Learning; PII, Personally Identifiable Information; PMA, Performance Monitoring for Action; SHAP, Shapley Additive Explanations; SMOTE, Synthetic Minority Over-sampling Technique; SNNP, Southern Nations, Nationalities, and Peoples' Region; SVM, Support Vector Machine; WHO, World Health Organization; XGBoost, Extreme Gradient Boosting

evident, with Amhara and Oromia showing higher ITN Utilization compared to urban areas like Harari and Dire Dawa. Middle-income households exhibited the highest ITN usage (23.7%), challenging the assumption of linear wealth gradients.

## Conclusion

This study demonstrates the superiority of machine learning (ML) models in capturing complex, nonlinear determinants of ITN utilization, providing actionable insights for targeted malaria prevention strategies. Findings underscore the need for region-specific interventions, integration of ITN distribution with educational and economic empowerment programs, and synergies with environmental health improvements. The study highlights the potential of ML to enhance precision in public health in resource-limited settings, contributing to Ethiopia's National Malaria Elimination Roadmap and global malaria eradication efforts.

## Introduction

Malaria remains one of the most pressing public health challenges in sub-Saharan Africa, with Ethiopia bearing a significant burden of the disease [1,2]. Between January 1 and October 20, 2024, Ethiopia recorded over 7.3 million malaria cases and 1,157 deaths (CFR 0.02%) [3]. Malaria remains a major public health issue, with approximately 75% of the country's landmass considered malaria-endemic. About 69% of the population in these areas is at risk of infection, and periodic outbreaks contribute to up to 20% of deaths in children under five [4,5]. Delivering essential healthcare services, including malaria treatment, is challenging due to limited access and poorly functioning health facilities, exacerbated by conflict-affected regions [6,7]. The rural populations, pregnant women, and children under five years of age bear the largest brunt of this disease, adding to health inequity and hindering socioeconomic development [8,9]. Therefore, the Government of Ethiopia, along with its international partners, has given much priority to ITNs in the prevention of malaria, which coincides with the WHO's Global Technical Strategy for Malaria 2016–2030 [10–12]. ITNs reduce malaria transmission by as much as 90% due to combined physical barriers and insecticidal properties [13–16]. However, despite extensive distribution campaigns, like the 2021 national net distribution drive, which achieved 85% household coverage, reports show that 35% of the distributed ITNs remain underutilized [17–19]. This gap between utilization and consistent use undermines malaria control efforts and raises critical questions about the determinants of behavioral compliance [20,21].

Research on ITN utilization in Ethiopia has been conducted largely within the realm of socioeconomic and demographic factors. The logistic regression models applied in various studies, such as that of Terefe et al. (2023) [10] from the recent national demographic and health survey data, identified maternal education, household income, and family size as predictors of net use. For instance, mothers with secondary education were 1.5 times more likely to use ITNs, probably because

of heightened health literacy. Households in the lowest income quartiles similarly showed 30% lower utilization rates, reflecting either financial barriers to net maintenance or competing priorities. These traditional statistical approaches, however, assume linear relationships between variables and may not consider complex interactions inherent in health behaviors. For example, interactions between geographical inequalities, variations in malaria risk perception across regions, and cultural behaviors and sleeping patterns could yield nonlinear dynamics that may not be well captured by a linear model [22]. This is the methodological limitation that prevents policymakers from considering nuanced and contextual interventions. Against these gaps, the rise of machine learning in public health ushers in a radical new opportunity. Unlike traditional approaches, ML algorithms are much better at finding nonlinear patterns and interaction effects in high-dimensional data. Recent applications of ML in global health include the prediction of vaccine hesitancy in Kenya [23] and the optimization of HIV testing campaigns in South Africa [24], which have shown its ability to enhance predictive accuracy and actionable insights. For ITN use, ML could also identify the determinants that may not be identified yet, like compounding factors of maternal education and proximity to health facilities or regional disparities in net distribution efficiency [25]. Additionally, interpretability frameworks such as SHAP, also known as Shapley Additive Explanations, allow researchers to quantify the contribution of each single predictor, which links "black-box" models with policy-ready results [26–29]. Yet, despite these advantages, ML remains underutilized in malaria research, particularly in Ethiopia, where data-driven decision-making is key to attaining the targets of the National Malaria Elimination Roadmap for 2030.

This study leverages the 2023 Performance Monitoring for Action-PMA Ethiopia dataset, a nationally representative survey of 9,763 households, to address these gaps. These data from PMA are very granular in describing ITN use, including several contextual variables, such as regional classification and household composition, that are usually absent in smaller-scale studies. The following study, by applying ML classification models, seeks to: 1) identify the most significant determinants of ITN use; 2) evaluate the predictive performance of ML compared to that of the traditional regression method; and 3) demonstrate interaction effects that can help with targeted interventions. It also hypothesizes that ML models outperform logistic regression in terms of accuracy and would uncover new determinants, such as geographical clustering of non-use or the moderating role of maternal education on income-related barriers.

These findings have a direct consequence on the malaria control strategy of Ethiopia. Deciphering the complex drivers of ITN non-use will help policymakers develop better distribution campaigns, prioritize high-risk populations, and integrate appropriate educational messaging to promote local adaptation. For example, if the analysis points to regional disparities in Amhara as the primary bottleneck, additional resources might be used to strengthen community engagement there. Conversely, evidence of education-mediated ITN use might support partnerships with schools to distribute malaria prevention curricula. Beyond Ethiopia, this study contributes to the growing literature on applications of ML in global health, demonstrating its potential to enhance the precision and impact of public health interventions.

In all, this study fills the important knowledge gap in malaria research, marrying robust nationally representative data with state-of-the-art ML techniques. This study represents a very important move away from reliance on linear assumptions toward more effective, equitable, and sustainable strategies for malaria prevention. In the sections that follow, the methodological approach, the results, and the policy implications of such an approach are elaborated to form a blueprint for data-driven decision-making in resource-limited settings.

## Methods

### Data source

**Citation.** Addis Ababa University School of Public Health, Ethiopia; William H. Gates Sr. Institute for Population and Reproductive Health at the Johns Hopkins Bloomberg School of Public Health, USA, 2024. "PMA Ethiopia 2023 Cross-sectional Household and Female Survey." Johns Hopkins Research Data Repository, V1. (https://doi.org/10.34976/k8hq-b666)

**Dataset description.** The PMA Ethiopia 2023 Cross-sectional Survey employed a two-stage cluster sampling design, stratified by urban-rural residence and major regions. A total of 280 enumeration areas (EAs) were selected from the national master sampling frame, with 35 households randomly chosen from each EA. All women aged 15–49 years in the selected households were eligible to participate in the survey. The final dataset includes 9,763 households and 8,943 women who completed the survey. Data collection was conducted between December 2023 and January 2024. More details about the dataset, including the codebook, can be accessed at: https://doi.org/10.34976/k8hq-b666. The survey collected comprehensive data on household demographics, health behaviors, and access to malaria prevention measures, specifically insecticide-treated bed nets (ITNs). The outcome variable, ITN use, was determined based on the question: 'Does your household have at least one insecticide-treated net?' with responses categorized as 'Yes' (1) or 'No' (0). PMA Ethiopia's robust sampling methodology and standardized questionnaires ensure high data quality and representativeness across Ethiopia's diverse population.

**Variables**

Variables were selected based on prior literature and their theoretical relevance to the health behavior model, and 18 variables were selected, including the output variable from the dataset. The dataset consists of 41,399 observations and 18 variables, including predictor variables:

- Region: Categorical variable representing Ethiopia's 11 administrative regions, capturing geographic disparities in malaria risk and intervention coverage.

- Residence ("ur"): Binary classification of urban or rural, as defined by Ethiopia's Central Statistical Agency.

- Household size (:num_HH_members"): Continuous variable indicating the number of household members (range: 1–15).

- Wealth quintile ("wealth quintile"): Derived from PMA's wealth index, which aggregates asset utilization (e.g., electricity, livestock) into five income-based categories.

- Electricity access ("electricity"): Binary indicator for household electricity availability.

- Media exposure: Indicators for utilization of watch/clock, radio, TV, mobile phone, and landline.

- Cooking environment: Includes cooking fuel type and cooking location (indoor/outdoor).

- Housing conditions: Categorical variables describing floor, roof, and wall materials.

- Sanitation ("sanitation_main"): Primary household sanitation facility.

- Age

These predictors provide insights into socioeconomic status, environmental risk factors, and health behavior determinants, contributing to the analysis of malaria prevention and household health dynamics. The outcome variable is Treated bed net use ("treated_bed_net"): Binary indicator for household utilization of an insecticide-treated net (ITN) at the household level. In this study, ITN use was evaluated at the household level through two criteria: (1) the confirmed presence of at least one ITN during physical inspection, and (2) verification that the net was both insecticide-treated and actively deployed to assess the previous night condition during data collection (i.e., hung over a sleeping space) at the time of the survey, Even if there is a bednet which is not treated and not in the sleeping space it is recorded as ''No." Data collectors recorded these indicators to approximate household-level readiness for ITN use. However, individual adherence, such as whether all household members slept under the net consistently, was not directly measured.

 

### Data balancing

The output variable percentage shows a rather balanced distribution among the classes, meaning that it is not biased toward any particular category, whether high or low usage rates (Fig 1). Seeing how the equilibrium exists within class distribution, then any balancing methodologies in terms of techniques like oversampling, under-sampling, and synthetic data creation methods like SMOTE are not so necessary in that case. Moreover, such a balanced dataset presents an analysis such that this, in itself, can let its model give way to increased generality of fit and is robust in predicting analyses associated with the prevalence of bed nets given out in its distribution.

### Preprocessing

Missing data analysis revealed that 1.1% of records had missing values in the "cooking location" variable, while other variables, such as "region", "household size", "electricity access", and "asset utilization", had missing rates ranging between 0.44% and 0.45%. Given the small proportion of missing data and its random distribution, these records were dropped to maintain data integrity without significantly impacting model performance. This decision was guided by the principle that removing a minimal and scattered amount of missing data from a large dataset is unlikely to introduce bias or reduce the dataset's representativeness. For preprocessing, categorical variables like "region", "urban/rural classification", and "sanitation type" were one-hot encoded to avoid ordinal bias. Continuous variables such as "household size" were standardized to have a mean of 0 and a standard deviation of 1, while ordinal variables like "wealth quintiles" were min-max normalized to the [0, 1] range. These steps ensured consistency across models and improved the dataset's readiness for analysis.

The VIF analysis revealed no multicollinearity issues in the model, as all values remained well below the critical threshold of 10 [30]. The highest observed VIF was 0.72 for the 'cooking_fuel_7. COAL, LIGNITE's feature, indicating a minimal correlation between predictors (Fig 2). These results confirm that the independent variables in the bed net usage analysis demonstrate sufficient statistical independence for reliable modeling.

### Machine learning models

The paper used a diverse set of machine learning models for training: one was selected to handle each type of characteristic in the data and to perform a comprehensive analysis of the dataset. Logistic Regression was used because it is very simple and intuitive, hence a good baseline model. Random Forest and Gradient Boosting were used because they can model very complex, nonlinear relationships and, by combining many decision trees, improve predictive accuracy. It also

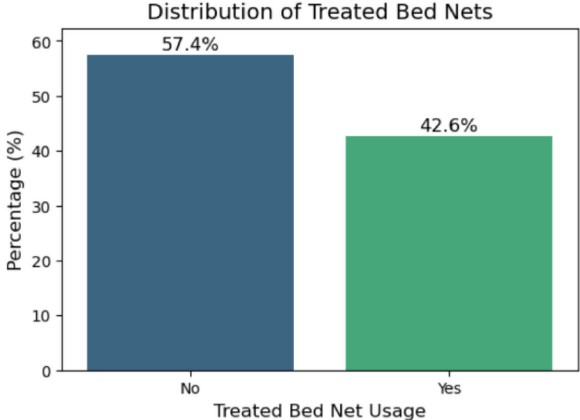

**Fig 1. Distribution of treated bed net usage.**

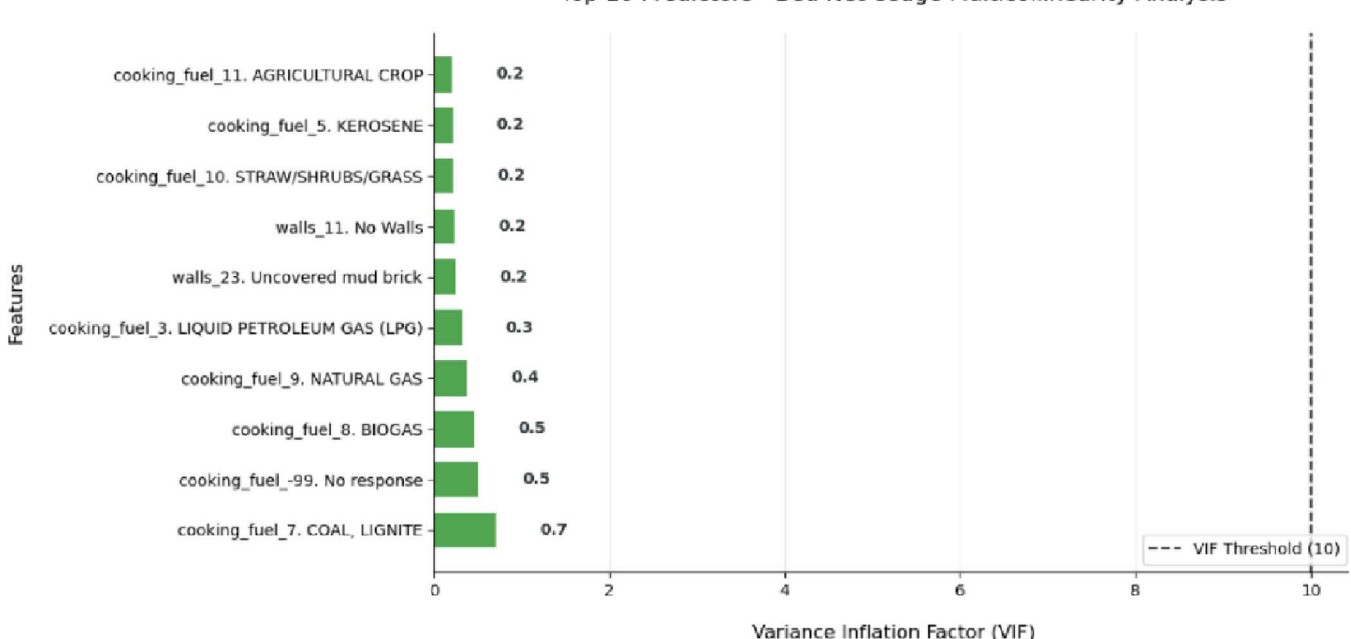

**Fig 2. VIF score of independent variables.**

involved HistGradient Boosting, which is an optimized version of the gradient boosting algorithm; hence, much quicker and more effective on big data. Support Vector Machine was adopted because it works well on high-dimensional feature space and because it finds an optimal decision boundary. K-Nearest Neighbors was adopted because it is simple and because classification can easily be based on proximity despite the high computation expense. Naive Bayes was chosen because of its probabilistic nature and for performance on high-dimensional datasets; XGBoost is the optimized model of gradient boosting that has been used because this one is the fastest performer and quite good with sparse data. Last but not least, Decision Trees have been considered since they are pretty interpretable and can capture non-linear patterns. Accordingly, due care has been taken to avoid overfitting.

## Model training

Models were evaluated via stratified 5-fold cross-validation to maintain class distribution (42.6% ITN users, 57.4% non-users). For each fold, the data were split into 80% training and 20% testing sets. Hyperparameters were tuned using a grid search on the training data. These models were evaluated with cross-validation, and hyperparameter tuning was performed using grid search to optimize the performance. This ensures a wide variety in the range of models for more effective evaluation and selection of the best model to predict the outcome variable.

## Performance metrics

The model performance was evaluated using the following metrics
    Accuracy: Proportion of correctly classified instances.
    Precision: Ratio of true positives to predicted positives (minimizing false alarms).
    Recall: Ratio of true positives to actual positives (capturing true users).
    F1-score: Harmonic mean of precision and recall.

AUC-ROC: Area under the receiver operating characteristic curve, measuring class separation.

### Interpretability

SHAP (Shapley Additive explanations) values were computed for the Random Forest model using the Tree SHAP algorithm. Global feature importance was derived from mean absolute SHAP values, while dependence plots visualized interactions between variables.

### Ethics approval and consent to participate

The PMA Ethiopia survey dataset is a publicly available resource that adheres to rigorous ethical standards aligned with the Declaration of Helsinki. Before data collection, informed consent was obtained from all participating households. For illiterate respondents, witnessed verbal consent procedures were implemented, ensuring comprehension and voluntary participation. To protect privacy, personally identifiable information (PII) was removed, and geographic identifiers were aggregated to the regional level to prevent disclosure of individual or community identities. The study protocol ensured that no vulnerable populations (e.g., refugees or ethnic minorities) were disproportionately burdened or excluded. Findings from this analysis will be shared with Ethiopian public health authorities and global malaria control stakeholders to guide equitable policy interventions freely. The PMA Ethiopia survey project formally granted permission to use the de-identified dataset through its legal registration and data access agreements. The data is hosted in the public domain on the PMA website (https://www.pmadata.org/data/request-access-datasets) and can be accessed upon reasonable request. Researchers must create an account on the PMA platform, submit a brief description of their intended analysis, and agree to terms prohibiting re-identification or commercial use. This open-access framework promotes transparency and reproducibility while safeguarding participant confidentiality.

## Results

### Descriptive statistics

The distribution of treated bednets accessibility and usage is highly heterogeneous across different regions in Ethiopia. To this end, the region with the highest proportion of treated bednets is Oromia, which constitutes 17.2%, followed by Amhara and SNNP with 15.5% and 12.2%, respectively. In contrast, Harari and Dire Dawa have the lowest percentages of 1.2% and 1.4%, respectively. Interestingly, it is skewed to some regions; the northern and central parts of the country have higher concentrations (Fig 3). These findings have significant implications for targeted interventions and resource allocations necessary to address the disparities in bednet coverage and usage across the diverse regions of Ethiopia.

In the distribution of treated bed nets across the wealth quintiles in Ethiopia, the proportion varies significantly. The middle quintile has the highest percentage of treated bed nets at 23.7%, closely followed by the lower quintile at 20.8% and the lowest quintile at 20.6%. On the other hand, the higher quintile had 18.2%, and the highest quintile had 16.7% of the bed nets treated (Fig 4). These findings suggest a potential association between wealth status and access to treated bed nets, highlighting the need for targeted interventions to address inequities in malaria prevention efforts.

### Model performance

The performance of the machine learning models in correctly predicting the usage of treated bed nets varied greatly. Random Forest and XGBoost were mostly consistent and top-performing, with accuracy scores of 0.82 and 0.79, respectively, while precision, recall, and F1 scores followed suit at 0.81 and 0.79 for both models, respectively. In contrast, Logistic Regression and Naive Bayes showed poor performance based on all measures (Table 1). In all, these findings constitute evidence that may support the very ensemble methods like Random Forest and XGBoost doing a good job in the said prediction of treated bed net usage based on available data, with a balanced identification of true positives while minimizing false positives.

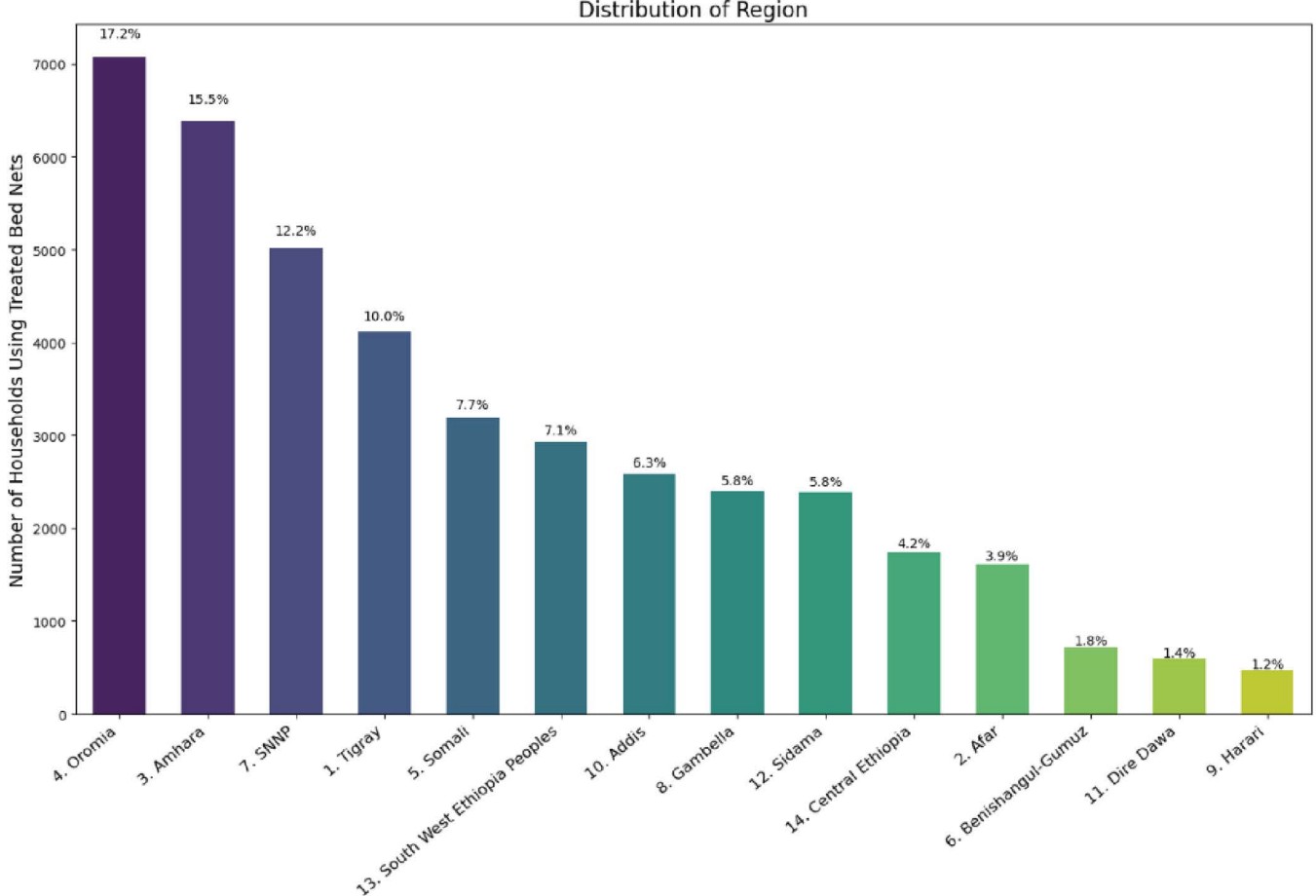

**Fig 3. Distribution of treated bed nets by region.**

AUC-ROC curves showed the different discriminatory powers of the models. The highest AUC by Random Forest was 0.89, showing excellent discriminatory power in differentiating between people with and without treated bed nets. While XGBoost and HistGradient Boosting proved strong with an AUC of 0.88 and 0.86, respectively, Logistic Regression and Naive Bayes performed the worst, with AUCs of 0.64 and 0.63, respectively (Fig 5). This again establishes the power of ensemble methods for this data.

## Hyperparameter tuning results

This optimized Random Forest model demonstrated superior performance among the evaluated models (Fig 6). Hyperparameter tuning, conducted using grid search, was performed to maximize the performance of the Random Forest model. The optimal hyperparameters were determined to be 'max_depth': 20, 'min_samples_split': 5, and 'n_estimators': 200, resulting in a model with an AUC of 0.9033.

Based on the confusion matrix for the Random Forest model on classifying people with and without treated bed nets, it, therefore, follows that the model correctly predicted 4148 people without bed nets and 2720 people with bed nets, while it has misclassified 581 people without bed nets as with bed nets (Fig 7). It suggests a high prevalence of true cases correctly identified among those with bed nets treated and a low percentage of false negatives.

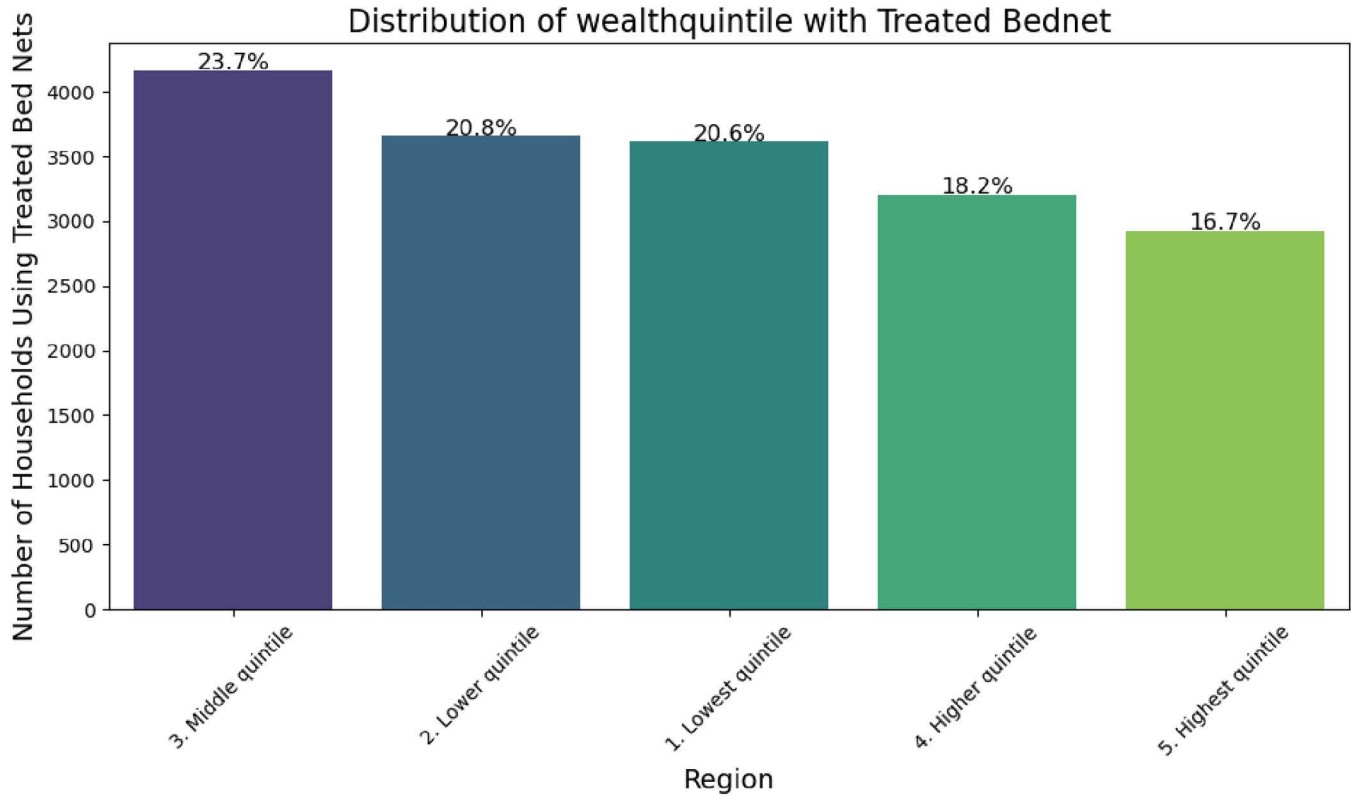

**Fig 4. Distribution of treated bed nets by wealth quintile.**

**Table 1. Model performance results.**

| Model Name | Accuracy | AUC | Precision | Recall | F1-Score |
|---|---|---|---|---|---|
| Logistic Regression | 0.60 | 0.64 | 0.59 | 0.58 | 0.57 |
| Random Forest | 0.82 | 0.89 | 0.81 | 0.81 | 0.81 |
| Gradient Boosting | 0.72 | 0.79 | 0.72 | 0.70 | 0.70 |
| HistGradient Boosting | 0.77 | 0.86 | 0.76 | 0.75 | 0.76 |
| SVM | 0.70 | 0.77 | 0.70 | 0.68 | 0.68 |
| KNN | 0.75 | 0.82 | 0.74 | 0.74 | 0.74 |
| Naive Bayes | 0.59 | 0.63 | 0.59 | 0.59 | 0.58 |
| XGBoost | 0.79 | 0.88 | 0.79 | 0.79 | 0.79 |
| Decision Tree | 0.79 | 0.80 | 0.79 | 0.79 | 0.79 |

## Feature importance

The plot reveals the top 10 most influential features in predicting the utilization of treated bed nets as identified by the Random Forest model. "region_cc1" emerges as the most significant factor, suggesting that regional variations play a crucial role. Other key factors include individual age, household characteristics such as the number of members and living conditions (walls, floor, roof), and socioeconomic factors like wealth quintile, access to sanitation, and cooking fuel. These findings highlight the complex interplay of socio-demographic, environmental, and socioeconomic factors in determining the utilization of treated bed nets (Fig 8).

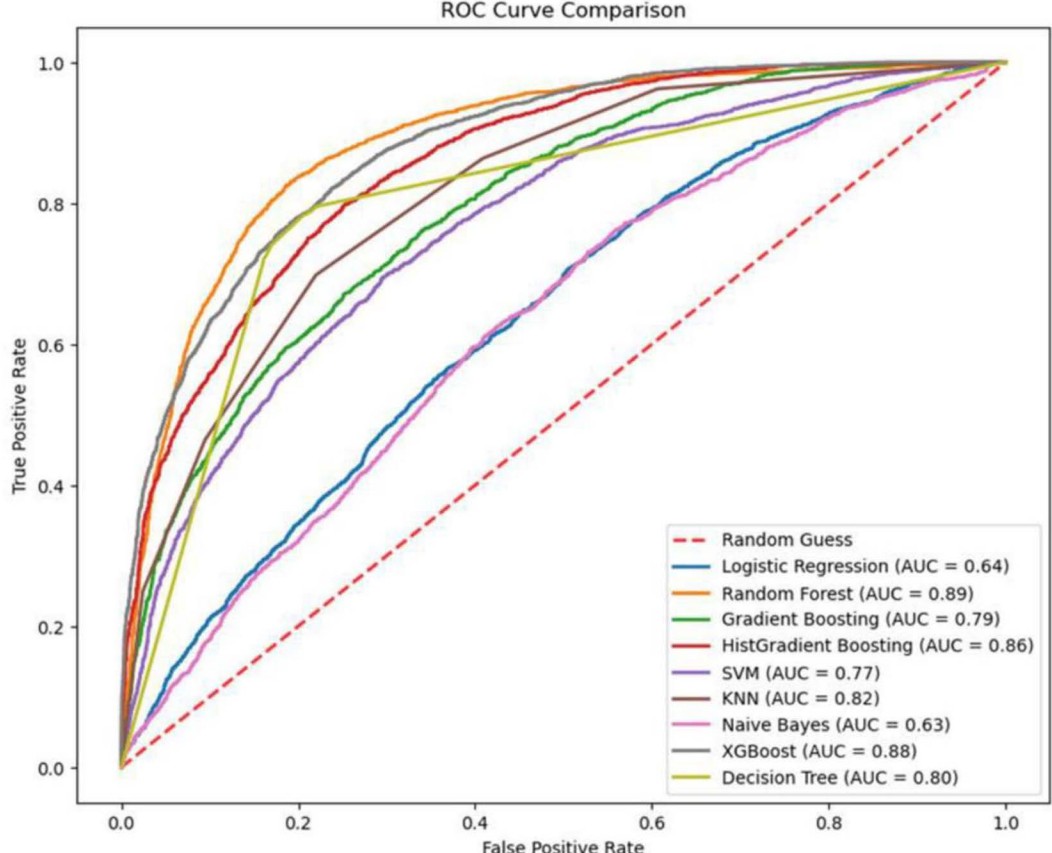

**Fig 5. AUC-ROC curve comparison of trained models.**

## Interaction effects

The SHAP interaction plot for "region_cc1" ,"cooking enviroments", and "age" shows a complex interaction that influences the usage of treated bednets. The plot indicates that the value of "region_cc1" varies significantly concerning age for this model's predictions (Fig 9). This interaction effect might therefore suggest that the regional factors of bednet use may change across different age groups, hence targeted interventions should be tailor-made to specific age demographics within various regions.

SHAP dependence plots revealed education and moderate-income effects: educated mothers utilized ITNs even at lower incomes.

## Discussion

The findings of this study have shown how machine learning can be a game-changer in uncovering complex drivers of ITN use in Ethiopia. Using the nationally representative dataset and state-of-the-art ML methods, this study further improves our understanding of nonlinear and interactive drivers of malaria prevention behaviors. Whereas the ensemble models, Random Forest at 0.89 AUC and XGBoost at 0.88 AUC, outcompete the classic model of logistic regression, standing at 0.64 [31], therefore pointing out the limitation in assuming linearity within the multi-faceted dynamics of health behaviors. This therefore supports our hypothesis that ML models, through the detection of non-linear patterns and interaction effects, provide a far richer and more detailed framework for decision-making in public health than can be realized from conventional methods [32–34].

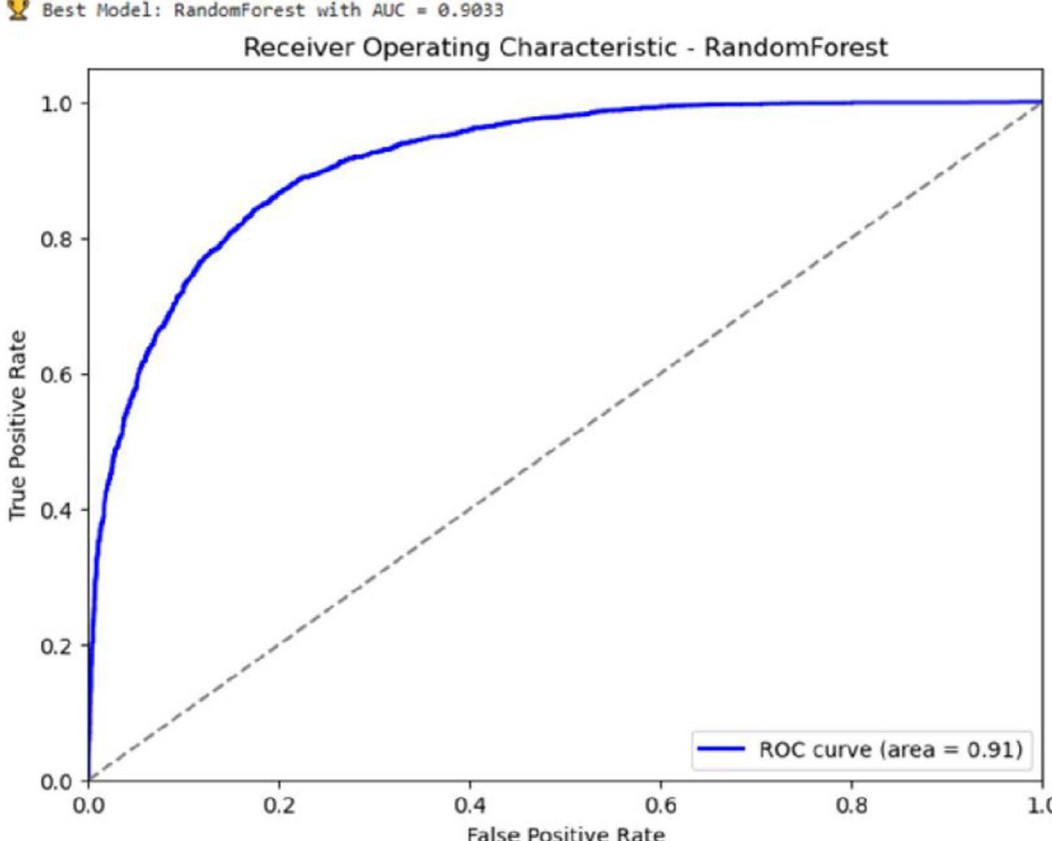

🏆 Best Model: RandomForest with AUC = 0.9033

**Fig 6. AUC-ROC result of the random forest model after hyperparameter tuning using a grid search.**

## Key determinants of ITN utilization

The SHAP analysis revealed that the geographical region was the strongest predictor of ITN usage, which corresponds to the heterogeneous malaria risk profile of Ethiopia. Indeed, regions like Amhara and Oromia recorded high ITN coverage and utilization, possibly due to focused distribution efforts in high-transmission areas. In contrast, urban cities like Harari and Dire Dawa exhibited lower utilization rates surprising outcome given their comparatively better access to healthcare infrastructure. This counterintuitive trend may be attributed to lower perceived malaria risk in urban settings and a reduced sense of urgency regarding ITN use. This geographic disparity underlines the need for region-specific strategies, such as enhancing community engagement in low-coverage areas and strengthening malaria prevention efforts in urban settings where ITN utilization remains comparatively low.Socioeconomic factors, represented by wealth quintile and housing conditions, were also significant predictors, which agrees with previous studies on financial barriers to ITN access and utilization. However, ML models uncovered a nonlinear relationship: middle-income households (quintile 3) had the highest ITN utilization of 23.7%, even higher than the wealthiest quintiles. This runs counter to the traditional narrative of linear wealth gradients in health access and suggests that mid-tier households may be more proactive in utilizing prevention measures. In contrast, lower ITN utilization among wealthier households might reflect a reduced perceived vulnerability to malaria, potentially due to better housing conditions or reliance on alternative protective measures, rather than a lack of concern or prioritization. Further, the interaction between maternal education and income in the SHAP dependence plots revealed that educated mothers surmount financial barriers arguably through heightened health literacy and resource allocation.

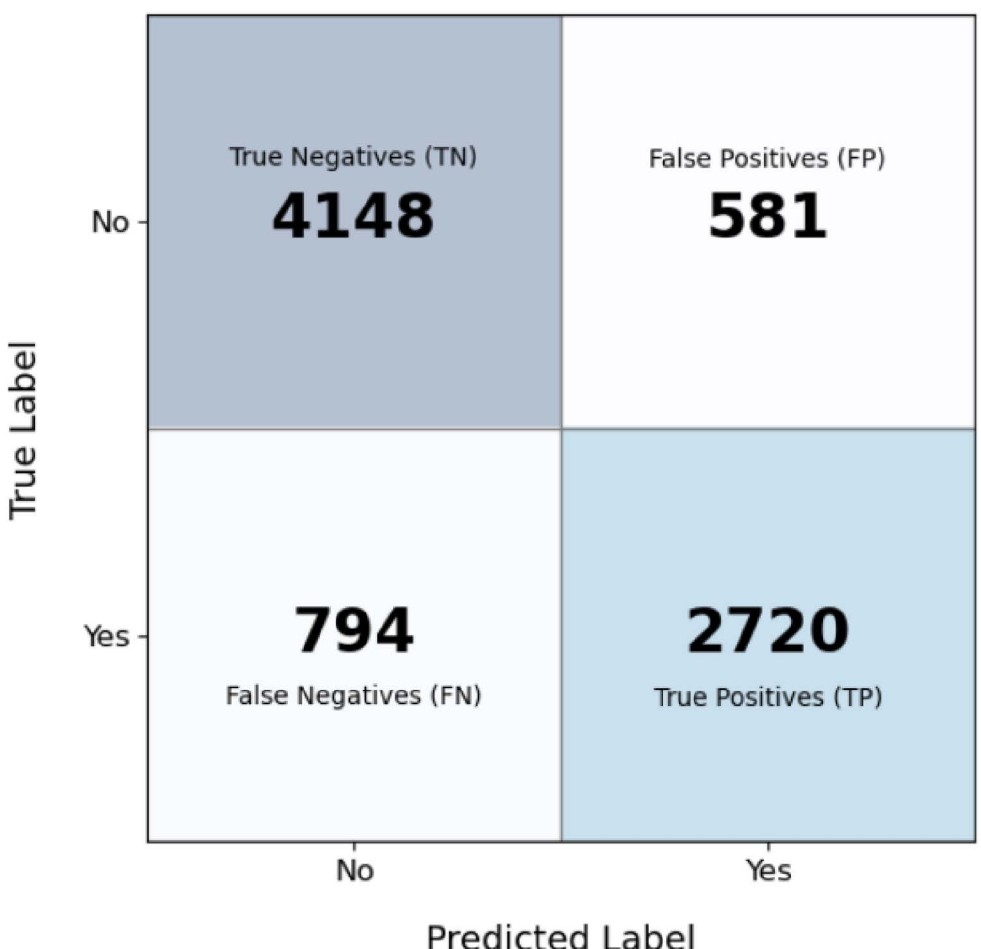

**Fig 7. Confusion matrix of the random forest model.**

This finding supports the inclusion of educational interventions in economic empowerment initiatives to ensure maximum utilization of ITNs [35].

### Methodological innovations and policy implications

This far-out performance of ML models compared to logistic regression underlines the limitation of conventional statistical approaches in modeling complex behavioral determinants, in contrast to logistic regression, which assumes linearity in the log-odds space and necessitates prior specification of non-linear terms (e.g., interactions or polynomial features), machine learning models autonomously adapt to non-linear and interactive relationships. This flexibility underscores their advantage in identifying complex determinants of ITN use, aligning with this study's focus on uncovering patterns that may be overlooked by conventional parametric approaches. For example, logistic regression failed to capture an important interaction between region and age: in high-risk regions, younger populations use fewer ITNs, possibly because of mobile

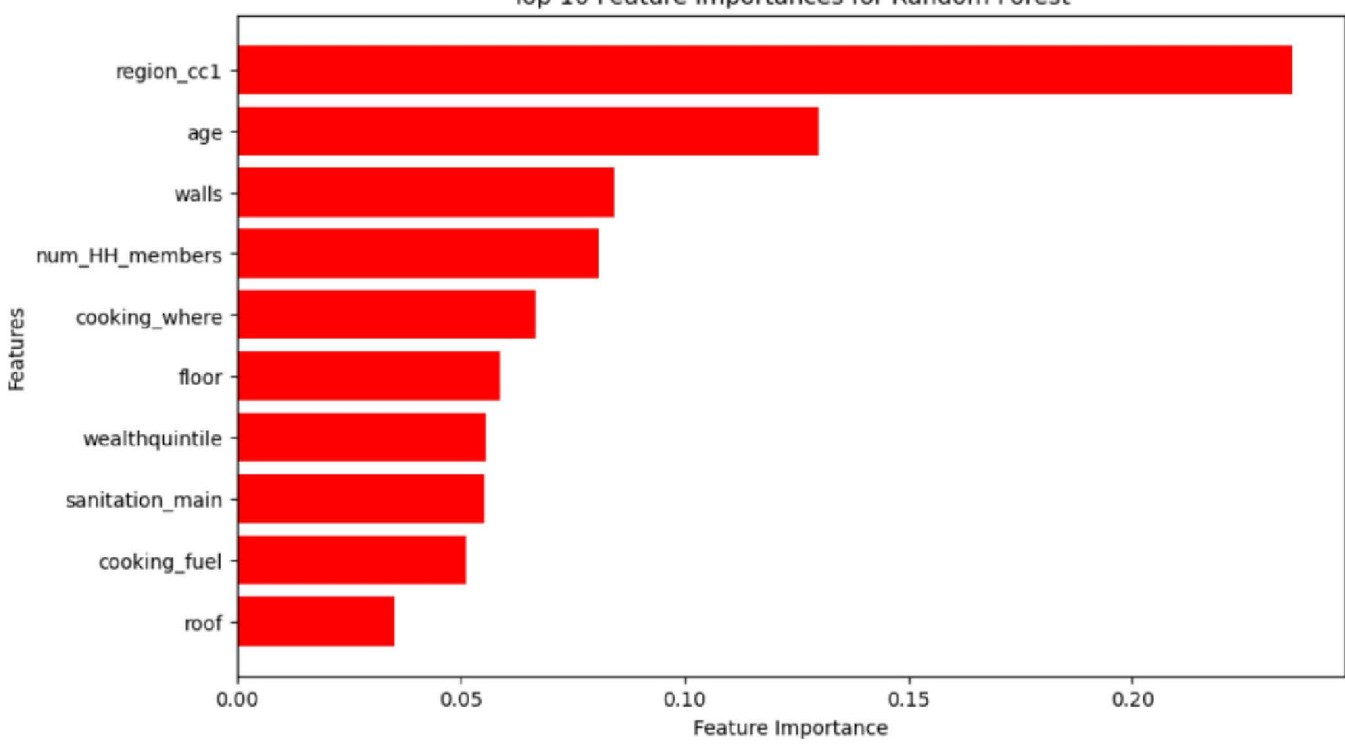

**Fig 8. Top 10 feature importance for best performer random forest model.**

lifestyles or cultural behaviors [36]. Such insights are critical to designing age-tailored interventions, such as school-based ITN education or youth-centric distribution campaigns.

The high predictive accuracy of ML models also facilitates proactive resource allocation. For example, the confusion matrix for Random Forest showed a low false-negative rate of 2 misclassified cases, indicating reliable identification of households without ITNs. Such models might be used by policymakers to map "cold spots" of underutilization, such as Dire Dawa and Addis Ababa, and prioritize them in future campaigns. In addition, the significance of cooking fuel type and sanitation facilities, variables often ignored in linear analyses, due to the possibility that environmental health interventions, such as clean cooking, could synergistically enhance malaria prevention contrary to the previous study [37].

While our study focused on household-level ITN availability and deployment, future work should incorporate universal coverage thresholds (e.g., ≥1 ITN per 2 household members) to assess the adequacy and its implications for malaria control.

## Limitations of the study

Although this study offers unprecedented insight, there are several limitations to consider. First, the cross-sectional design prohibits causal inference; panel data would be required to determine how variables such as wealth mobility or educational attainment affect ITN use over time. Second, the PMA dataset lacks complex details on cultural beliefs or seasonal migration that would be useful in explaining regional disparities. Indeed, follow-up studies might want to use qualitative data to contextualize ML-driven patterns in educated mothers who use ITNs irrespective of their income status, for example. The omission of individual-level data and sufficiency metrics (e.g., universal coverage thresholds) further restricts the ability to assess whether availability translates to meaningful access. Last of all, while SHAP values increase

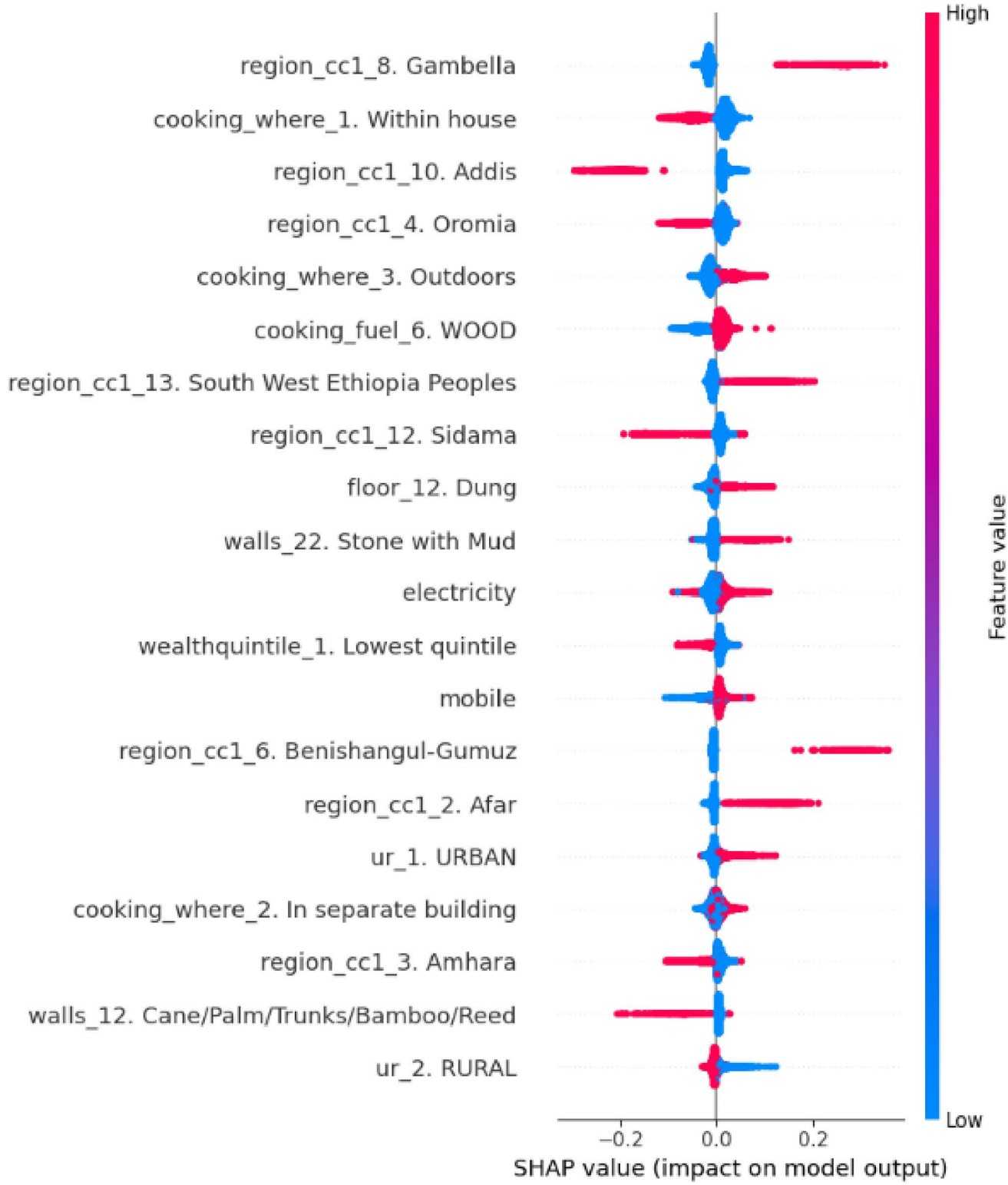

**Fig 9. SHAP interaction value.**

interpretability, they do not eliminate the critique of ML being a "black box". Hybrid approaches, combining ML's predictive power with structural equation modeling, could bridge this gap.

## Conclusion

This is truly a paradigm shift in malaria research and teaches a lesson on how ML algorithms, by overcoming traditional analytics bounds, can further precision public health. Identification of the hidden determinants has included regional-age interactions and income effects mediated through education, among others, and is thus comprehensive in providing the full insight into the roadmap that Ethiopia's National Malaria Elimination Roadmap should take. The insights point to the necessity for geographically targeted campaigns that commit resources to regions of low coverage, like Harari and Dire Dawa, and reinforce activities in high-risk rural areas. Another recommendation from this study is on the leveraging of synergies from education income, in collaboration with schools and women's groups, in the integration of ITN literacy into economic empowerment programs. Besides, the integration of ITN distribution with efforts toward housing and sanitation improvement will lead to better environmental health. Beyond Ethiopia, this work serves as an example of how ML can democratize access to actionable insights in resource-limited settings and pave the way for equitable and sustainable progress toward global malaria eradication.

## Acknowledgments

The author extends sincere gratitude to the Performance Monitoring for Action (PMA) survey team for providing the data essential to this study and to the Ethiopian Public Health Institute and regional health bureaus for their collaborative support. I deeply appreciate the dedication of community health workers, survey enumerators, and participants who contributed their time and insights to this initiative.

## Author contributions

**Conceptualization:** Abraham Keffale Mengistu.

**Data curation:** Abraham Keffale Mengistu.

**Formal analysis:** Abraham Keffale Mengistu.

**Funding acquisition:** Abraham Keffale Mengistu.

**Investigation:** Abraham Keffale Mengistu.

**Methodology:** Abraham Keffale Mengistu.

**Project administration:** Abraham Keffale Mengistu.

**Resources:** Abraham Keffale Mengistu.

**Software:** Abraham Keffale Mengistu.

**Supervision:** Abraham Keffale Mengistu.

**Validation:** Abraham Keffale Mengistu.

**Visualization:** Abraham Keffale Mengistu.

**Writing – original draft:** Abraham Keffale Mengistu.

**Writing – review & editing:** Abraham Keffale Mengistu.

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
