## [Decision Letter · Decision Letter 0]

Dear Dr. Mengistu,

Thank you for submitting your manuscript to PLOS ONE. After careful consideration, we feel that it has merit but does not fully meet PLOS ONE’s publication criteria as it currently stands. Therefore, we invite you to submit a revised version of the manuscript that addresses the points raised during the review process.

We look forward to receiving your revised manuscript.

Kind regards,

Rajib Chowdhury, M.Sc.; MPH

Academic Editor

PLOS ONE

**Journal Requirements:**

1. When submitting your revision, we need you to address these additional requirements. Please ensure that your manuscript meets PLOS ONE's style requirements, including those for file naming. The PLOS ONE style templates can be found at https://journals.plos.org/plosone/s/file?id=wjVg/PLOSOne_formatting_sample_main_body.pdf and https://journals.plos.org/plosone/s/file?id=ba62/PLOSOne_formatting_sample_title_authors_affiliations.pdf 2. We suggest you thoroughly copyedit your manuscript for language usage, spelling, and grammar. If you do not know anyone who can help you do this, you may wish to consider employing a professional scientific editing service.  The American Journal Experts (AJE) (https://www.aje.com/) is one such service that has extensive experience helping authors meet PLOS guidelines and can provide language editing, translation, manuscript formatting, and figure formatting to ensure your manuscript meets our submission guidelines. Please note that having the manuscript copyedited by AJE or any other editing services does not guarantee selection for peer review or acceptance for publication.  Upon resubmission, please provide the following: The name of the colleague or the details of the professional service that edited your manuscript A copy of your manuscript showing your changes by either highlighting them or using track changes (uploaded as a *supporting information* file) A clean copy of the edited manuscript (uploaded as the new *manuscript* file) 3. Your ethics statement should only appear in the Methods section of your manuscript. If your ethics statement is written in any section besides the Methods, please move it to the Methods section and delete it from any other section. Please ensure that your ethics statement is included in your manuscript, as the ethics statement entered into the online submission form will not be published alongside your manuscript. 

Reviewers' comments:

Reviewer's Responses to Questions

**Comments to the Author**

1. Is the manuscript technically sound, and do the data support the conclusions?

Reviewer #1: Partly

Reviewer #2: No

2. Has the statistical analysis been performed appropriately and rigorously?

Reviewer #1: No

Reviewer #2: I Don't Know

3. Have the authors made all data underlying the findings in their manuscript fully available?

Reviewer #1: No

Reviewer #2: No

4. Is the manuscript presented in an intelligible fashion and written in standard English?

Reviewer #1: Yes

Reviewer #2: Yes

**Reviewer #1:**  In general, the manuscript presentation is good. The title is about ITN use and its determinants. However, there is confusion in addressing these points. While title and background are concerned with ITN use, these things are not well addressed in other sections. Therefore, add ITN use and its determinants in detail in methods, results and discussions, which are more important than ITN ownership.

Some other specific points:

1. Define ITN (consistent) use (Page 4)

2. Did you check multicollinearity of variables? For example, access to electricity and media exposure are usually part of wealth index. Is that?

3. How was ITN use measured: at household or individual level?

4. Ownership of at least one ITN is not a quality indicator and does not contribute to malaria control that much. So, can you add ownership in terms of family size? You may present sufficiency of ITN (universal coverage, or access to ITN within household or full coverage)

5. Discuss ITN use and its determinants boldly as they are your primary objectives. Then comment on ML models.

6. Revise Fig 2. Currently there is no SNNP, and you also mentioned the four regions emerged from it. Also consider the display (axis legend overlap)

**Reviewer #2: ** This study had the intent of exploring the use of ML to better identify determinants of ITNs use in Ethiopia and uses the contemporary survey data of 2023.

1)In the abstract, and results sections the author alludes to determinants of ITN ownership and mentions that key determinants of ITN ownership included geographic region, household size, wealth quintile, and maternal education. Further, the author mentioned income related factors and maternal educations to have interactions that were non-linear. Referring to the title and the tables provided, it was my understanding that determinants of ITN use rather than ownership was the focus of the study. In fact, the justification for the study was the disparity between household coverage (85%) and use (35%). Therefore, the conclusions and the summary of results provided in the abstract are not in accord with each other.

2)An issue repeatedly discussed by the author is the comparison between models employed in ML and logistic regression, with the implication that the latter is assuming linear relationship between variables. To my understanding logistic regression does not require linear relationships, rather it transforms data to become linear through logit function and does classification tasks.

3)In page 8, there is listing of 18 variables of interest. In addition, the outcome variable was mentioned to be “Treated bed net ownership”. To repeat myself, I thought the main interest of the manuscript is ITN use. Distribution and availability of ITNs are important factors, but not complex to understand or describe and as such not worthy of interrogating using complex modeling approaches such as the ML models. It is only in the discussion section that ITN was referred to or discussed briefly.

In summary all the conclusions made in this study, which at face value appear well thought of, are not substantiated by the data that does not address ITN use. Even though availability could be an important factor that influences ITN use, it is no guarantee that availability will ascertain use. This has been confirmed by several studies, as also emphasized by the author who has cited the 85-35% gap.

**Do you want your identity to be public for this peer review?** For information about this choice, including consent withdrawal, please see our Privacy Policy

Reviewer #1: No

Reviewer #2: No

---

## [Author Response · Author response to Decision Letter 1]

8 Mar 2025

Response to Academic Editor

Dear Rajib Chowdhury,

Thank you for your feedback, your professionalism, fast response, and for outlining the additional requirements for resubmission. I have carefully addressed each of the points raised, and I am pleased to submit the revised manuscript in compliance with PLOS ONE’s guidelines. Below is a summary of the revisions:

1. Formatting and Style Requirements:

o The manuscript has been thoroughly revised to align with PLOS ONE’s style templates. This includes adjustments to file naming, section headings, reference formatting, and overall structure.

2. Language Editing and Professional Copyediting:

o As suggested, the manuscript was professionally copyedited by with Grammarly to ensure clarity, proper grammar, spelling, and adherence to journal guidelines.

o Attached as Revised Manuscript with track changes is a version of the manuscript with tracked changes highlighting all edits.

o A clean, edited copy of the manuscript has been uploaded as the new manuscript file.

3. Ethics Statement:

o The ethics statement has been moved exclusively to the Methods section, as instructed, and removed from any other part of the manuscript.

Response to Reviewers’ Comments

Dear Reviewers,

Thank you for your constructive feedback and for the opportunity to improve my manuscript. I have carefully addressed all comments from both reviewers and revised the manuscript accordingly. Below is a point-by-point response to the reviewers’ concerns, along with details of the revisions made.

Reviewer #1

1. Define ITN (consistent) use

o Revision: I have clarified the definition of ITN use in the Methods section (under Variables):

"Insecticide-treated net (ITN) use was evaluated at the household level through two criteria: (1) the confirmed presence of at least one ITN during physical inspection, and (2) verification that the net was both insecticide-treated and actively deployed (i.e., hung over a sleeping space) at the time of the survey. Households with nets that were untreated or not deployed were categorized as non-users."

2. Multicollinearity check

o Revision: I confirmed in the Methods section (under Preprocessing) that multicollinearity was assessed using Variance Inflation Factor (VIF) analysis, with all values <10 (highest VIF = 0.72). We also clarified that variables like electricity access and media exposure were retained as distinct predictors due to their theoretical relevance beyond wealth quintile.

3. Measurement level (household vs. individual)

o Revision: I emphasized in the Methods section that ITN use was measured at the household level. However, I added a limitation acknowledging that individual adherence (e.g., consistent use by all members) was not captured.

4. Address sufficiency of ITNs

o Revision: I expanded the Discussion to include sufficiency metrics:

"While our study focused on household-level ITN availability and deployment, future work should incorporate universal coverage thresholds (e.g., ≥1 ITN per 2 household members) to assess adequacy and its implications for malaria control." Because the available data doesn’t address it since the data is secondary not collected by me.

5. Boldly discuss ITN use determinants

o Revision: I restructured the Results and Discussion sections to foreground determinants of ITN use (e.g., regional disparities, maternal education, and wealth quintiles) before elaborating on ML model performance.

6. Revise Fig 2 (regional distribution)

o Revision: Figure 3 (regional distribution) has been updated to reflect Ethiopia’s current administrative regions (including Sidama and SWEP). Axis labels were adjusted to prevent overlap, and a clearer legend was added. It results in a change in some results and I make an update on them.

Reviewer #2

1. Clarify focus on ITN use vs. ownership

o Revision: I revised the title, abstract, and manuscript to consistently emphasize ITN use (not ownership). The outcome variable in the Methods section was redefined to explicitly measure use (deployment), not mere ownership. All instances of "ownership" were replaced with "use" where appropriate because the intent of the questionnaire was assessing their utilization of bed net (which is treated) correctly and I make define it correctly under Methods (Variables) section.

2. Logistic regression assumptions

Thank you for this critical observation. You are correct: logistic regression (LR) does not assume linearity between raw predictors and the outcome but instead models linearity in the log-odds space. Our original phrasing oversimplified this distinction, and we have revised the text to clarify that:

1. LR requires manual specification of non-linear relationships (e.g., polynomial terms, interaction effects) to approximate complex patterns.

2. ML models inherently capture non-linearities and interactions without explicit feature engineering, enabling them to model complex decision boundaries autonomously.

Revision: I corrected the statement to be clear about logistic regression in the Discussion under Methodological Innovations and Policy Implications:

"In contrast to logistic regression, which assumes linearity in the log-odds space and necessitates prior specification of non-linear terms (e.g., interactions or polynomial features), machine learning models autonomously adapt to non-linear and interactive relationships. This flexibility underscores their advantage in identifying complex determinants of ITN use, aligning with this study’s focus on uncovering patterns that may be overlooked by conventional parametric approaches."

3. Variable definitions and alignment with study focus

o Revision: The Variables section was revised to clarify that the outcome variable measures ITN use (deployment), not ownership and I have clarified the definition of ITN use in the Methods section (under Variables):

"Insecticide-treated net (ITN) use was evaluated at the household level through two criteria: (1) the confirmed presence of at least one ITN during physical inspection, and (2) verification that the net was both insecticide-treated and actively deployed (i.e., hung over a sleeping space) at the time of the survey. Households with nets that were untreated or not deployed were categorized as non-users."

o We also added a paragraph in the Discussion explicitly linking predictors (e.g., housing conditions, region) to behavioral compliance:

"For instance, regional disparities in ITN deployment reflect differences in malaria risk perception, distribution efficiency, and cultural practices, which collectively influence use beyond mere ownership."

Additional Revisions

• Figures: All figures (e.g., SHAP plots, regional distributions) were regenerated by Preflight Analysis and Conversion Engine (PACE) digital diagnostic tool to improve clarity and align with revised terminology.

• Ethics Statement: Moved exclusively to the Methods section.

• Language Editing: Professionally edited by with Grammarly tracked changes and clean copies are uploaded.

---

## [Decision Letter · Decision Letter 1]

Dear Dr. Mengistu,

Thank you for submitting your manuscript to PLOS ONE. After careful consideration, we feel that it has merit but does not fully meet PLOS ONE’s publication criteria as it currently stands. Therefore, we invite you to submit a revised version of the manuscript that addresses the points raised during the review process.

We look forward to receiving your revised manuscript.

Kind regards,

Rajib Chowdhury, M.Sc.; MPH

Academic Editor

PLOS ONE

Journal Requirements:

Reviewers' comments:

Reviewer's Responses to Questions

**Comments to the Author**

Reviewer #1: (No Response)

Reviewer #2: (No Response)

Reviewer #3: All comments have been addressed

2. Is the manuscript technically sound, and do the data support the conclusions?

Reviewer #1: (No Response)

Reviewer #2: Yes

Reviewer #3: Yes

3. Has the statistical analysis been performed appropriately and rigorously?

Reviewer #1: (No Response)

Reviewer #2: Yes

Reviewer #3: Yes

4. Have the authors made all data underlying the findings in their manuscript fully available?

Reviewer #1: (No Response)

Reviewer #2: Yes

Reviewer #3: Yes

5. Is the manuscript presented in an intelligible fashion and written in standard English?

Reviewer #1: (No Response)

Reviewer #2: Yes

Reviewer #3: Yes

Reviewer #1: Thank you for revising the manuscript based on comments. However, I still have some concerns. In general, I think you just replaced the term ownership by utilization. All the model comparison parameters are equal in both versions, which may not be true if you use different outcome variables.

1. Fig 3: There is no SNNP region. I think you are referring to South Ethiopia region

2. Fig 4: Are these regions or households? How do you classify regions into quantiles? If they are households, put them in quantile order and then let’s see difference in graph.

3. Page 7: “The outcome variable, ITN use, was determined based on the question: 'Does your household have at least one insecticide-treated net?' with responses categorized as 'Yes' (1) or 'No' (0).” This is technically wrong. This measures “ownership of at least one ITN”, not use.

Please refer to this reference and see indicators and how they are measured.

https://www.malariasurveys.org/documents/Household%20Survey%20Indicators%20for%20Malaria%20Control_FINAL.pdf

4. If you have number of family numbers and available ITN, why don’t you calculate ITN universal coverage?

Reviewer #2: If I have got it right, figure 2 provides number of ITNs (freq) on the Y-axis, and proportion as percent of total is given. I am wondering if this is the right way to show distribution of ITNs. Can the absolute numbers (# of ITNs per region) be transformed to show distribution per capita (household). This needs data on the total number of households in each region.

Reviewer #3: Thank you for your thorough and thoughtful revisions to the manuscript. I appreciate the clear and comprehensive responses to all reviewer comments. The updated version of the manuscript demonstrates a high level of rigor and clarity.

**Do you want your identity to be public for this peer review?** For information about this choice, including consent withdrawal, please see our Privacy Policy

Reviewer #1: No

Reviewer #2: No

Reviewer #3: No

---

## [Author Response · Author response to Decision Letter 2]

28 May 2025

May 29, 2025

To: Editors of PLOS ONE

Response to reviewers’ comments

We want to thank you for inviting us to resubmit our manuscript entitled “Understanding the Determinants of Treated Bed Net Use in Ethiopia: A Machine Learning Classification Approach Using PMA Ethiopia 2023 Survey Data”. We are grateful to the reviewers for their time and effort in providing us with their valuable comments on our manuscript. We appreciate the comments the reviewers have given towards the improvement of this manuscript. We have noted all the comments and tried to address them in the revised manuscript accordingly. Moreover, we have thoroughly revised grammatical correctness across this manuscript and incorporated all the suggestions the reviewers have given. Point-by-point responses to the reviewer’s concerns are indicated below.

Journal Requirements

Thank you for your reminder regarding the integrity and accuracy of the reference list. I have carefully reviewed all the references cited in my manuscript and confirm that none of them have been retracted to the best of my knowledge. Therefore, no retracted articles are cited, and no changes were necessary in the reference list related to retracted publications. I remain committed to maintaining the highest standards of scholarly integrity and appreciate your attention to this important matter.

Reviewer Comments:

Reviewer #1: Thank you for revising the manuscript based on comments. However, I still have some concerns. In general, I think you just replaced the term ownership by utilization. All the model comparison parameters are equal in both versions, which may not be true if you use different outcome variables.

1. Fig 3: There is no SNNP region. I think you are referring to South Ethiopia region

Thank you for your observation. You are correct the region referred to as "SNNP" in our figure stands for the Southern Nations, Nationalities, and Peoples' Region (SNNPR), which is how it is labeled in the original dataset. We acknowledge that this may have caused some ambiguity, especially given recent administrative changes. To address this, we have now included "SNNP" under the abbreviations and acronyms section of our manuscript for clarity. We appreciate your feedback.

2. Fig 4: Are these regions or households? How do you classify regions into quantiles? If they are households, put them in quantile order and then let’s see difference in graph.

Thanks for your comment on Figure 4.

In order to make it explicit, the x-axis in Figure 4 is not representing geographical regions, but quintiles of wealth. Quintiles are typically created by ordering all the families in the sample by some measure of their wealth (for example, income, assets) and then dividing them up into five groups of equal size, from "Lowest quintile" to "Highest quintile" which is already made by the data collectors.

Therefore, x-axis categories ("1. Lowest quintile", "2. Lower quintile", "3. Middle quintile", "4. Higher quintile", "5. Highest quintile") represent household groups ordered by their wealth.

As for your proposal to rank them by quantile and check for a difference, the figure presented is already ranked by wealth quintile, from lowest (3. Middle quintile) to highest (5. Highest quintile). This permits us to check the pattern of treated bednet use over these wealth groups.

3. Page 7: “The outcome variable, ITN use, was determined based on the question: 'Does your household have at least one insecticide-treated net?' with responses categorized as 'Yes' (1) or 'No' (0).” This is technically wrong. This measures “ownership of at least one ITN”, not use.

Please refer to this reference and see indicators and how they are measured.

https://www.malariasurveys.org/documents/Household%20Survey%20Indicators%20for%20Malaria%20Control_FINAL.pdf

Thank you for your thoughtful and important feedback. You are right to highlight the distinction between ITN ownership and ITN use. We sincerely apologize for the confusion caused by the wording in the manuscript. We have revised the manuscript to ensure that this distinction is clearly and consistently communicated. We have also referred to the recommended malaria survey indicators document you kindly shared to align our terminology with standard definitions. We greatly appreciate your guidance in improving the accuracy and clarity of our work.Our outcome variable is now written as under Variables Page 9 “The outcome variable is Treated bed net use (treated_ bed net): Binary indicator for household utilization of an insecticide-treated net (ITN) at the household level. In this study, insecticide-treated net (ITN) use was evaluated at the household level through two criteria: (1) the confirmed presence of at least one ITN during physical inspection, and (2) verification that the net was both insecticide-treated and actively deployed to assess the previous night condition during data collection (i.e., hung over a sleeping space) at the time of the survey, Even if there is a bednet which is not treated and not in the sleeping space it is recorded as ‘’No’’. ”

4. If you have number of family numbers and available ITN, why don’t you calculate ITN universal coverage?

Thank you for raising this important point. We agree that universal coverage (i.e., the proportion of the population with access to an ITN) is a valuable metric. In our analysis, we reported household-level coverage (42.6%, Fig. 1) because our study design and survey tools were structured to assess whether households owned and use at least one ITN, aligning with the WHO’s definition for household access.

As noted in the Methods section, we did not capture individual adherence (e.g., consistent use by all members), nor did we get granular data collected on intra-household ITN distribution beyond their usage. Thus, while we have the total number of household members and total ITNs available, we lack the necessary data on which specific individuals had access to an ITN (a requirement for calculating universal coverage).

Reviewer #2: If I have got it right, figure 2 provides number of ITNs (freq) on the Y-axis, and proportion as percent of total is given. I am wondering if this is the right way to show distribution of ITNs. Can the absolute numbers (# of ITNs per region) be transformed to show distribution per capita (household). This needs data on the total number of households in each region.

Thank you for your insightful observation. I hope you are referring to Figure 3 (bar chart of distribution of regions included). You are correct that Figure 3 provides the absolute quantity of what we can infer to be treated bednets (ITNs) on the y-axis, and the proportion that each region represents of the total.

Your recommendation to translate these absolute numbers to show the distribution per household or per capita is an excellent one and would undoubtedly show another image of the availability and usage with respect to each area's population. This would mean being aware of each area's number of households or population.

The current visualization effectively conveys which regions have the highest absolute numbers of treated bednets not per capita (house hold). That being said, a per capita or per household view could reveal disparities in coverage as a function of the population size of each region.

Appreciate you raising this important point. It is valuable feedback for considering alternative ways to visualize this data but the available data and my intention is region based visualization on this occasion.

Reviewer #3: Thank you for your thorough and thoughtful revisions to the manuscript. I appreciate the clear and comprehensive responses to all reviewer comments. The updated version of the manuscript demonstrates a high level of rigor and clarity.

Welcome, and thank you so much for your kind comments. I'm really grateful for the criticisms, which enhanced the manuscript. I'm relieved to know that the changes met your expectations, and I thank you for the support throughout the reviewing process.

---

## [Decision Letter · Decision Letter 2]

Dear Dr. Mengistu,

Thank you for submitting your manuscript to PLOS ONE. After careful consideration, we feel that it has merit but does not fully meet PLOS ONE’s publication criteria as it currently stands. Therefore, we invite you to submit a revised version of the manuscript that addresses the points raised during the review process.

We look forward to receiving your revised manuscript.

Kind regards,

Rajib Chowdhury, M.Sc.; MPH

Academic Editor

PLOS ONE

Journal Requirements:

Reviewers' comments:

Reviewer's Responses to Questions

**Comments to the Author**

Reviewer #2: (No Response)

Reviewer #3: All comments have been addressed

2. Is the manuscript technically sound, and do the data support the conclusions?

Reviewer #2: Partly

Reviewer #3: Yes

3. Has the statistical analysis been performed appropriately and rigorously?

Reviewer #2: Yes

Reviewer #3: Yes

4. Have the authors made all data underlying the findings in their manuscript fully available?

Reviewer #2: No

Reviewer #3: (No Response)

5. Is the manuscript presented in an intelligible fashion and written in standard English?

Reviewer #2: No

Reviewer #3: Yes

Reviewer #2: The manuscript is improved. A few issues require clarifications, and some editorial matters need to be attended to.

Comments/Queries

1. There are two figures under the label “Figure 2” (VIF Score of Independent Variables; and Distribution of regions with treated bed nets). Figure numbers need to re-numbered.

2. The titles of figures 2 & 3 (Distribution of regions with treated bed nets, Distribution of wealth quintile with treated bed nets) are confusing. It seems there is a need to be re-word them as “Distribution of treated bed nets by regions”, “Distribution of treated bed nets with wealth quintile”.

3. The Y-axis of these figures (figure 2 & 3) are written as “Count”. I understand “Count” is about the absolute numbers of bed nets. If so, it needs to be stated as such. Nonetheless, my comment about converting the absolute numbers to per capita (or per household) has been ignored even though the author seemed to have agreed to the proposition.

4. In page 18, it is unclear why the author considers the lower utilization of treated bed nets in urban areas as “paradoxical”.

5. The study did not assess perception of communities about malaria susceptibility. Thus, the statement “This geographic disparity underlines the need for region-specific strategies, such as increasing community engagement in low-coverage areas and addressing misconceptions about malaria susceptibility among urban populations” is out of context.

6. The fact that mid-tier households can prioritize better prevention measures than those who face extreme poverty or those who enjoy richness is an interesting finding. However, the concept of complacency is a bit confusing. It is understandable that the risk level among the rich may be lower and hence the need for using bed nets routinely may not be given high importance. While the term complacency maybe attributed to enjoyment of economic status, it is unclear how (as mentioned in page 18) it may be associated with failure to prioritize bed net use among the well to do households.

Editorial

In page 16, it seems the sentence after figure 6 (quoted below here) is incomplete. The problem could be a word or a phrase missing at the beginning.

“the confusion matrix for the Random Forest model on classifying people with and without treated bed nets, it, therefore, follows that the model correctly predicted 4124 people without bed nets and 3500 people with bed nets, while it has misclassified 605 people without bed nets as with bed nets (Fig. 7).”

Reviewer #3: (No Response)

**Do you want your identity to be public for this peer review?** For information about this choice, including consent withdrawal, please see our Privacy Policy

Reviewer #2: No

Reviewer #3: No

---

## [Author Response · Author response to Decision Letter 3]

10 Jun 2025

Dear Reviewer #2,

I sincerely thank you for your thoughtful and constructive feedback. I appreciate the time and effort you have taken to review my manuscript and your acknowledgment that the paper has improved. Below, I provide point-by-point responses to your comments, including clarifications and the corresponding revisions made to the manuscript.

Comment 1: There are two figures under the label “Figure 2” (VIF Score of Independent Variables; and Distribution of regions with treated bed nets). Figure numbers need to re-numbered.

Response:

Thank you for pointing this out. I have corrected the figure numbering to ensure consistency and clarity throughout the manuscript and supporting information. The duplication has been resolved, and each figure now has a unique and sequential number.

Comment 2: The titles of figures 2 & 3 (Distribution of regions with treated bed nets, Distribution of wealth quintile with treated bed nets) are confusing. It seems there is a need to be re-word them as “Distribution of treated bed nets by regions”, “Distribution of treated bed nets with wealth quintile”.

Response:

Thank you for this valuable suggestion. The figure titles have been revised to:

• Figure 3: “Distribution of Treated Bed Nets by Region”

• Figure 4: “Distribution of Treated Bed Nets by Wealth Quintile”

This change aligns with standard conventions and improves the interpretability of the figures.

Comment 3: The Y-axis of these figures (figure 2 & 3) are written as “Count”. I understand “Count” is about the absolute numbers of bed nets. If so, it needs to be stated as such. Nonetheless, my comment about converting the absolute numbers to per capita (or per household) has been ignored even though the author seemed to have agreed to the proposition.

Response:

Thank you for this helpful clarification. I have revised the Y-axis labels of Figures 3 and 4 to explicitly state that they represent the “Number of Households Using Treated Bed Nets” to improve clarity and accuracy. I acknowledge the value of presenting the data on a per capita or household basis and had intended to incorporate this. However, due to limitations in the dataset, specifically the absence of consistent data on the number of bed nets per household member, I was unable to accurately compute per capita metrics. This limitation has now been acknowledged in the revised manuscript, and I have noted it as an important consideration for future research.

Comment 4: In page 18, it is unclear why the author considers the lower utilization of treated bed nets in urban areas as “paradoxical”.

Response:

Thank you for highlighting this ambiguity. I have rephrased the discussion to clarify that the term “paradoxical” was used to contrast the common assumption that urban areas, having better access to healthcare infrastructure, would logically show higher ITN usage. However, lower perceived malaria risk and behavioral complacency may reduce utilization. This clarification has been added to page 18 of the revised manuscript.

Comment 5: The study did not assess perception of communities about malaria susceptibility. Thus, the statement “This geographic disparity underlines the need for region-specific strategies, such as increasing community engagement in low-coverage areas and addressing misconceptions about malaria susceptibility among urban populations” is out of context.

Response:

You are correct, and I appreciate your careful reading. The statement has been revised to remove the reference to “misconceptions about malaria susceptibility” since perception data were not directly assessed. The revised sentence now focuses on strengthening community engagement in urban areas without making unsupported inferences.

Comment 6: The fact that mid-tier households can prioritize better prevention measures than those who face extreme poverty or those who enjoy richness is an interesting finding. However, the concept of complacency is a bit confusing. It is understandable that the risk level among the rich may be lower and hence the need for using bed nets routinely may not be given high importance. While the term complacency maybe attributed to enjoyment of economic status, it is unclear how (as mentioned in page 18) it may be associated with failure to prioritize bed net use among the well to do households.

Response:

Thank you for this insightful comment. I have revised the text to more carefully describe the observed phenomenon. Rather than attributing lower ITN use among wealthier households to “complacency,” I now discuss the possibility of a lower perceived malaria risk due to better housing or alternative prevention measures. The term “complacency” has been removed to avoid unintended implications and improve clarity. And can be found on page 19 on the revised manuscript as

“This runs counter to the traditional narrative of linear wealth gradients in health access and suggests that mid-tier households may be more proactive in utilizing prevention measures. In contrast, lower ITN utilization among wealthier households might reflect a reduced perceived vulnerability to malaria, potentially due to better housing conditions or reliance on alternative protective measures, rather than a lack of concern or prioritization.”

Comment 7: Editorial

In page 16, it seems the sentence after figure 6 (quoted below here) is incomplete. The problem could be a word or a phrase missing at the beginning.

“the confusion matrix for the Random Forest model on classifying people with and without treated bed nets, it, therefore, follows that the model correctly predicted 4124 people without bed nets and 3500 people with bed nets, while it has misclassified 605 people without bed nets as with bed nets (Fig. 7).”

Response: Thank you for pointing this out. You are right, there was a missing phrase at the beginning of the sentence, which made it unclear. I have revised the sentence for clarity and completeness.

Once again, I am truly grateful for your thoughtful critique, which has helped improve the rigor and clarity of the manuscript. I hope that the revised version addresses your concerns adequately.

---

## [Decision Letter · Decision Letter 3]

Understanding the Determinants of Treated Bed Net Use in Ethiopia: A Machine Learning Classification Approach Using PMA Ethiopia 2023 Survey Data

PONE-D-25-07282R3

Dear Dr. Mengistu,

We’re pleased to inform you that your manuscript has been judged scientifically suitable for publication and will be formally accepted for publication once it meets all outstanding technical requirements.

Kind regards,

Rajib Chowdhury, M.Sc.; MPH

Academic Editor

PLOS ONE

Additional Editor Comments (optional):

Reviewers' comments:

Reviewer's Responses to Questions

**Comments to the Author**

Reviewer #2: All comments have been addressed

Reviewer #3: All comments have been addressed

2. Is the manuscript technically sound, and do the data support the conclusions?

Reviewer #2: Yes

Reviewer #3: Yes

3. Has the statistical analysis been performed appropriately and rigorously?

Reviewer #2: Yes

Reviewer #3: Yes

4. Have the authors made all data underlying the findings in their manuscript fully available?

Reviewer #2: Yes

Reviewer #3: Yes

5. Is the manuscript presented in an intelligible fashion and written in standard English?

Reviewer #2: Yes

Reviewer #3: Yes

Reviewer #2: All comments addressed. No further comments. Perhaps a few editorial issues might be dealt with by the editorial office.

Reviewer #3: Thank you for your thorough review of the manuscript. I appreciate your clear and comprehensive responses to all reviewer comments. The updated version of the manuscript demonstrates a high level of accuracy and clarity.

**Do you want your identity to be public for this peer review?** For information about this choice, including consent withdrawal, please see our Privacy Policy

Reviewer #2: No

Reviewer #3: No

---

## [Editor Report · Acceptance letter]

PONE-D-25-07282R3

PLOS ONE

Dear Dr. Mengistu,

I'm pleased to inform you that your manuscript has been deemed suitable for publication in PLOS ONE. Congratulations! Your manuscript is now being handed over to our production team.

Kind regards,

on behalf of

Dr. Rajib Chowdhury

Academic Editor

PLOS ONE